# Vitamin D Deficiency in Cushing’s Disease: Before and After Its Supplementation

**DOI:** 10.3390/nu14050973

**Published:** 2022-02-25

**Authors:** Valentina Guarnotta, Francesca Di Gaudio, Carla Giordano

**Affiliations:** 1Department of Health Promotion, Maternal-Infantile Care, Excellence Internal and Specialist Medicine “G. D’Alessandro” [PROMISE], Section of Endocrine Disease and Nutrition, University of Palermo, 90127 Palermo, Italy; valentina.guarnotta@unipa.it; 2Biochemistry Head CQRC Division (Quality Control and Biochemical Risk), Department of Health Promotion, Maternal-Infantile Care, Excellence Internal and Specialist Medicine “G. D’Alessandro” [PROMISE], University of Palermo, 90127 Palermo, Italy; francesca.digaudio@unipa.it

**Keywords:** glucocorticoid, hypercortisolism, 25-hydroxyvitamin D, cholecalciferol

## Abstract

Background: The primary objective of the study was to assess serum 25-hydroxyvitamin D [25(OH)D] values in patients with Cushing’s disease (CD), compared to controls. The secondary objective was to assess the response to a load of 150,000 U of cholecalciferol. Methods: In 50 patients with active CD and 48 controls, we evaluated the anthropometric and biochemical parameters, including insulin sensitivity estimation by the homeostatic model of insulin resistance, Matsuda Index and oral disposition index at baseline and in patients with CD also after 6 weeks of cholecalciferol supplementation. Results: At baseline, patients with CD showed a higher frequency of hypovitaminosis deficiency (*p* = 0.001) and lower serum 25(OH)D (*p* < 0.001) than the controls. Six weeks after cholecalciferol treatment, patients with CD had increased serum calcium (*p* = 0.017), 25(OH)D (*p* < 0.001), ISI-Matsuda (*p* = 0.035), oral disposition index (*p* = 0.045) and decreased serum PTH (*p* = 0.004) and total cholesterol (*p* = 0.017) values than at baseline. Multivariate analysis showed that mean urinary free cortisol (mUFC) was independently negatively correlated with serum 25(OH)D in CD. Conclusions: Serum 25(OH)D levels are lower in patients with CD compared to the controls. Vitamin D deficiency is correlated with mUFC and values of mUFC > 240 nmol/24 h are associated with hypovitaminosis D. Cholecalciferol supplementation had a positive impact on insulin sensitivity and lipids.

## 1. Introduction

Vitamin D is the precursor of a hormone with pleiotropic effects. Its deficiency has been largely investigated and shown to be associated with many complications including diabetes mellitus, adrenal insufficiency, cardiovascular disease, neurological disorders and other endocrinopathies [1,2,3].

Vitamin D, also known as cholecalciferol, is first formed in the skin by the photolysis of 7-dehydrocholesterol and after hydroxylated in the liver to 25-hydroxyvitamin D [25(OH)D]. It is further transformed in the kidney into 1,25-dihydroxyvitamin D3 (1,25(OH)2D3) (calcitriol) that is the active form [4].

Cushing’s disease (CD) is characterized by a cortisol excess due to autonomous pituitary ACTH secretion. Patients with CD show many comorbidities such as cardiovascular disease, metabolic disease, diabetes mellitus, metabolic syndrome, dyslipidemia, obesity, osteoporosis/osteopenia and infections that contribute to increasing the mortality risk for these patients [5,6,7,8,9,10,11]. Indeed, GCs are key regulators of intermediary metabolism promoting hepatic gluconeogenesis and glycogenosis and on lipid metabolism favouring the deposition of fat to the upper trunk and the face [12]. They stimulate water diuresis, glomerular filtration rate and renal plasma flow and these effects result in arterial hypertension and atherosclerosis. GCs reduce bone remodelling, augment urinary calcium excretion and decrease the intestinal calcium absorption. In addition, they act on immune and hematological systems inhibiting the secretion of interleukins and increasing the red blood cell count, respectively [12]. 

An interesting relationship exists between glucocorticoids (GCs) and vitamin D values [13,14,15,16]. Indeed, exogenous steroid therapy has been reported to be associated with vitamin deficiency [13]. The mechanism by which GCs reduce 25(OH)D levels is not direct, but indirect, regulating vitamin D receptor expression in many tissues and cells [17,18]. Some authors have shown that treatment with dexamethasone in mice was associated with a decrease in 1α-hydroxylase which is involved in the conversion from 25(OH)D3 to the active metabolite 1,25(OH)2D3 and an increase in 24-hydroxylase, able to break down the active form of calcitriol, in inactive, reducing circulating 25(OH)D levels [19]. In a clinical setting, controversial data have been reported on GCs effects on serum 1,25(OH)2D concentrations [20,21,22,23]. A likely reason for these discrepancies might be the marked heterogeneity of the studied groups. Some of these studies were performed in humans [23,24,25,26], and others in animal models [27,28], but only a few studies were conducted in subjects with endogenous hypercortisolism.

Low serum 25(OH)D levels have significant skeletal and extra-skeletal consequences such as myopathy, high risk of fractures and also affect the immune system and metabolism. All of these systems are impaired in patients with hypercortisolism and a vitamin D deficiency may provide a further aggravation of CD comorbidities. Indeed, it may cause a reduced intestinal calcium absorption resulting in secondary hypocalcemia and hyperparathyroidism leading to a bone demineralization. Its deficiency can contribute to obesity and metabolic syndrome due to the lack of antiadipogenic effect of vitamin D and to cardiovascular disease by a deregulation of the renin–angiotensin–aldosterone system, cardiac contractility and increase in cytokine release [29]. In the end, vitamin D deficiency causes impaired insulin sensitivity and immune system [30]. 

The discrepancies that emerge in the above-mentioned studies suggest a need to investigate the role of 25(OH)D in patients with CD. Therefore, the primary objective of the study was to evaluate serum 25(OH)D levels in patients with CD, compared to a control group matched for age, BMI and gender, and search for a possible correlation with the degree of hypercortisolism. The secondary objective was to evaluate the response to a course of 150,000 U of cholecalciferol on metabolic and hormonal parameters 6 weeks after the administration in patients with CD. 

## 2. Materials and Methods

### 2.1. Subjects and Study Design

Fifty patients with active CD, 43 of them women (86%) and 7 of them men (20%) (mean age 50.9 ± 17.4 years; mean duration of disease 32.5 ± 22.4 years), followed from January 2016 to December 2020, by the Endocrinology of the University of Palermo, were included in the current study. Clinical practice guidelines and a recent consensus statement were used to diagnose CD [31,32].

We recruited a control group matched for age, BMI and gender in the same temporal period. It was composed of 48 patients, 33 women (82.5%) and 7 men (17.5%) (mean age 48.5 ± 13.4 years) were evaluated by our team for a suspicion not biochemically confirmed of Cushing’s syndrome (CS). 

In all patients, we evaluated phenotypic characteristics including moon face, facial rubor, dorsal fat pad or buffalo hump, defined as a fatty tissue deposit between the shoulders, purple striae, defined as wide, reddish-purple streaks, and myopathy defined as muscle weakness at the proximal level. 

We also assessed cardiovascular, metabolic and bone comorbidities. The diagnosis of metabolic syndrome was based on National Cholesterol Education Program Adult Treatment Panel (NCEP ATP III) criteria, while the diagnosis of diabetes mellitus and prediabetes were based on the American Diabetes Association (ADA, Arlington, VA, USA) criteria [33,34]. 

Among patients with diabetes mellitus (18 out of 50), 16 were treated with metformin alone, while 2 were treated with a combination of metformin and GLP-1 agonist receptors. Metformin and GLP-1 agonist receptors were discontinued 24 h and 2 weeks before metabolic evaluations, respectively, to avoid any interference with metabolic parameters. Diabetic patients were on good metabolic control (HbA1c ≤ 7%). Both CD patients and the controls were naïve to cholecalciferol.

In CD and the controls, BMI and waist circumference (WC), fasting serum lipids (total cholesterol (TC), HDL cholesterol, LDL cholesterol and triglycerides (TG), HbA1c, glycaemia, insulinaemia, albumin corrected calcium, phosphorus and parathyroid hormone (PTH) were assessed. To avoid seasonal influences, serum 25(OH)D levels were only assayed between winter and spring seasons (November–April). We evaluated urinary free cortisol (UFC) as the mean of three 24 h urine collections (mUFC), cortisol after a low dose of dexamethasone suppression test and plasma ACTH. We defined patients with mild hypercortisolism when mUFC levels not exceeding twice the upper limit of normal (ULN), moderate hypercortisolism by a level of mUFC more than 2 to 5 times the ULN and severe hypercortisolism by a mUFC level more than 5 times the ULN, as previously reported [35]. 

As defined by the Endocrine Society guidelines, we considered 25(OH)D deficiency for values < 20 ng/mL (50 nmol/L), insufficiency as levels of 20–30 ng/mL (50–75 nmol/L) and sufficiency for values ≥ 30 ng/mL (≥75 nmol/L) [36]. In addition, severe 25(OH)D deficiency was defined by levels < 10 ng/mL (<25 nmol/L) [37].

As markers of insulin sensitivity, we calculated the homeostatic model of insulin resistance (HOMA2-IR) [38], and in 32 patients with CD and in 40 controls who had no previous diagnosis of diabetes, we also evaluated the Matsuda index of insulin sensitivity (ISI-Matsuda) [39], the oral disposition index (DIo) [40] and the area under the curve for insulin (AUC2h insulinemia) and glucose (AUC2h glycaemia).

At the baseline visit, we assessed patients’ lifestyle habits: physical activity level, balanced diet (consumption of dairy products, meat, coffee, soft drinks), exposure to ultraviolet (UV) radiation, smoking status and alcohol use. 

We excluded patients with adrenal-dependent hypercortisolism, pregnancy, taking oral contraceptives, liver or renal disease, cholecalciferol supplementation within 3 months before the study, malabsorption syndrome and exposure to ultraviolet (UV) radiation (solarium and sunscreen usage).

Patients with CD received an oral load dose of cholecalciferol of 150,000 UI [41,42] and biochemical parameters (metabolic and hormonal) were assayed 6 weeks after administration.

The study protocol was approved by the Ethics Committee of the Policlinico Paolo Giaccone hospital. All patients signed a written informed consent.

### 2.2. Assays

Biochemical parameters were measured by standard methods (Modular P800, Roche, Milan, Italy), as previously reported [9]. 

Hormonal parameters were measured by electrochemiluminescence immunoassay (ECLIA, Elecsys, Roche, Milan, Italy) following the manufacturer’s instructions, as previously reported [9]. 

Mean UFC was measured by mass spectrometry, as previously reported [35]. 

Normal values for hormonal markers were defined as follows: ACTH 2.2–14 pmol/L and UFC 59–378 nmol/24 h. 

### 2.3. Statistical Analysis

We used statistical Packages for Social Science SPSS version 19 (SPSS, Inc., Chicago, IL, USA) for data analysis. The normality of quantitative variables was tested with the Shapiro–Wilk test. We calculated mean ± SD for continuous variables and rates and proportions for categorical variables. The differences between paired continuous variables (CD vs. controls) were analysed using one-way ANOVA. We used univariate Pearson correlation to evaluate the relations with the outcome parameters. For those variables which were significant at univariate correlation, we performed multiple linear regression analysis to identify independent predictors of the dependent variable 25(OH)D. A *p*-value of 0.05 was considered statistically significant. A receiver operating characteristic (ROC) analysis was performed to investigate the diagnostic ability of significantly associated risk factors to predict 25(OH)D deficiency. The ROC curve is plotted as sensitivity versus 1-specificity. The area under the ROC curve (AUC) was estimated to measure the overall performance of the predictive factors for serum 25(OH)D deficiency.

## 3. Results

At baseline, patients with CD had a higher frequency of arterial hypertension (*p* = 0.009), osteoporosis/osteopenia (*p* = 0.002), hypercholesterolemia (*p* = 0.002), diabetes mellitus (*p* = 0.026), myopathy (*p* < 0.001), facial rubor (*p* = 0.005), buffalo hump (*p* = 0.002) and hypovitaminosis deficiency (*p* = 0.001) than the controls (Table 1). 

By contrast, the controls had a higher frequency of vitamin D sufficiency (*p* = 0.004). Patients with CD also had higher WC (*p* = 0.031), PTH (*p* = 0.003), glycaemia (*p* = 0.010), HbA1c (*p* = 0.004), total cholesterol (*p* < 0.001), LDL cholesterol (*p* = 0.002), ACTH (*p* < 0.001), mUFC (*p* = 0.001), cortisol after a low dose of dexamethasone suppression test (*p* = 0.001) and lower 25(OH)D (*p* < 0.001), ISI-Matsuda (*p* = 0.007) and DIo (*p* = 0.003) than the controls (Table 2).

Six weeks after cholecalciferol treatment, patients with CD showed increased serum calcium (*p* = 0.017), 25(OH)D (*p* < 0.001), ISI-Matsuda (*p* = 0.035), DIo (*p* = 0.045) and a decrease in PTH (*p* = 0.004) and total cholesterol (*p* = 0.017) levels than at baseline (Table 3). 

Considering the degree of hypercortisolism, in patients with severe hypercortisolism we observed 25(OH)D deficiency in 73.1% of cases (53.8% of them had a severe deficiency), insufficiency in 12.5% of cases and sufficiency in 6.3% of cases. In patients with moderate hypercortisolism, we observed 25(OH)D deficiency in 64.7% of cases (29% of them had a severe deficiency), insufficiency in 23.5% of cases and sufficiency in 11.8% of cases. In patients with mild hypercortisolism, we observed deficiency in 52.9% of cases (20% of them had a severe deficiency), insufficiency in 41.1% of cases and sufficiency in 6% of cases.

At univariate correlation, in patients with CD at baseline, serum 25(OH)D was inversely correlated with glycaemia (r = −0.385, *p* = 0.019), HbA1c (r = −0.391, *p* = 0.017), WC (r = −0.373, *p* = 0.023), mUFC (r = −0.466, *p* = 0.033) and cortisol after a low dose of dexamethasone suppression test (r = −0.299, *p* = 0.049) (Table 4). In the controls, at baseline, 25(OH)D was inversely correlated with WC (r = −0.130, *p =* 0.042) (Table 4). 

Multivariate analysis showed that mUFC was independently inversely associated with 25(OH)D (*p* = 0.010) in patients with CD (Figure 1). In the controls, no significant associations were found. 

The ROC analysis showed that a cut-off of mUFC > 240 nmol/24 h was associated with 25(OH)D deficiency with a specificity of 100% and a sensitivity of 56.9%, AUC 0.803 (Figure 2).

## 4. Discussion

The present study shows that patients with active CD have lower serum 25(OH)D values than the controls and that serum 25(OH)D levels are inversely correlated with mUFC in CD. In addition, a cholecalciferol load is associated after 6 weeks from the administration with an improvement of serum 25(OH)D and glycometabolic and lipid parameters in patients with CD. Furthermore, we found that higher values of mUFC than 240 nmol/24 h are predictive of 25(OH)D deficiency. The degree of hypercortisolism evaluated by UFC levels is a useful parameter to quantify the “amount” of cortisol secretion, even though it is not sufficiently exhaustive to assess the aggressiveness of the disease [35]. Indeed, a combination of several factors, including the degree of hypercortisolism, but also the duration of the disease, age and other individual predisposing factors, contribute to the aggressiveness of the disease.

Long-standing studies were conducted on vitamin D levels in patients with CD. Patients with CD, with and without osteopenia, were compared before and after oral calcium load showing that serum 1,25 (OH)2D3 plasma levels were higher in subjects with osteopenia than in those without it, likely due to a secondary increase in PTH levels as an effect of hypercortisolism [19]. Another study investigated the effect of hypercortisolism and eucortisolism, showing a reduction in serum 25(OH)D levels, but not in 1,25 (OH)2D3 in patients with hypercortisolism. By contrast, two other studies found normal serum 25(OH)D values in patients with CD [23,24]. However, all the above-mentioned studies were conducted on a small sample of patients. Recently, a meta-analysis conducted on the studies that evaluated serum 25(OH)D levels in patients treated with GCs reported lower serum 25(OH)D levels in these patients compared to healthy subjects [16]. A hypothetical reason was that patients with CD had low 24-hydroxylase levels than the controls, causing an alteration of vitamin D catabolism. 

An interesting in vitro study in NCI-H295R cells found that treatment with 1,25(OH)2D3 decreased corticosterone secretion without affecting cortisol levels [43].

As expected, in the current study, we showed that treatment with cholecalciferol is associated with an improvement in insulin sensitivity and total cholesterol values in patients with CD. Indeed, cholecalciferol supplementation has been reported to be associated with improved peripheral insulin sensitivity and secretion in patients at high risk of diabetes or with type 2 diabetes [44]. A recent meta-analysis on 41 randomized controlled studies showed a significant improvement in total cholesterol levels after cholecalciferol supplementation. In addition, this improvement was more pronounced in patients with vitamin D deficiency [45,46]. 

A recent study compared the metabolism of vitamin D in patients with CD and controls after cholecalciferol treatment, showing that patients with CD had a higher 25(OH)D/24,25(OH)2D ratio than healthy controls, likely due to a decrease in 24-hydroxylase activity. The authors concluded that this alteration of vitamin D catabolism might have an influence on the effectiveness of cholecalciferol therapy in CD [47].

There are some limitations in the current study. First, the study is not randomized. Second, the dose of cholecalciferol administered is the same independently of the baseline serum 25(OH)D values. Third, we did not register the intake of milk and dairy products of the patients included in the study. 

In conclusion, serum 25(OH)D levels are lower in subjects with active CD compared to controls matched for age, BMI and gender. Vitamin D deficiency is correlated with mUFC and values of mUFC > 240 nmol/24 h are predictive of 25(OH)D deficiency. In addition, cholecalciferol supplementation has a positive impact on insulin sensitivity and lipids and therefore should be considered part of the treatment of patients with CD at diagnosis, in order to improve the comorbidities. However, further studies are needed to evaluate a possible effect of cholecalciferol supplementation on the aggressiveness of CD. 

## Figures and Tables

**Figure 1 nutrients-14-00973-f001:**
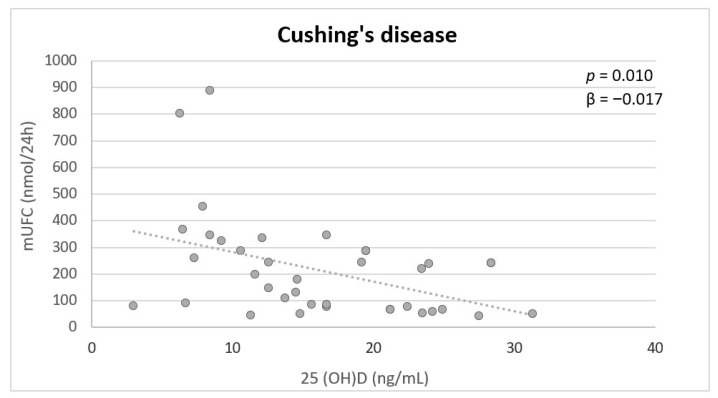
Independent variables associated with serum 25(OH)D in patients with active CD at multivariate analysis. mUFC: mean urinary free cortisol.

**Figure 2 nutrients-14-00973-f002:**
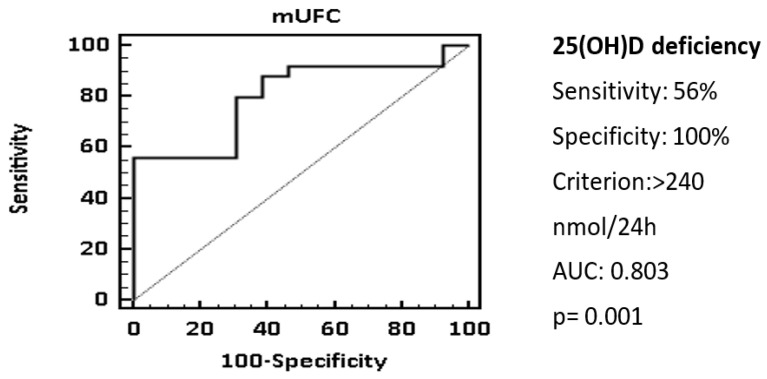
25(OH)D status and mUFC. ROC curve showed that a cut-off of mUFC > 240 nmol/24 h could be associated with 25(OH)D deficiency. Statistical analysis was performed using the chi-square test and receiver operator characteristic (ROC) curve analysis.

**Table 1 nutrients-14-00973-t001:** Comorbidities of patients with CD and controls at baseline.

	Controls	Cushing’s Disease	*p*
	(No. = 48)	(No. = 50)	
	Subjects (%)	Subjects (%)	
Gender			
Male	9 (18.7%)	7 (14%)	0.475
Female	39 (81.3%)	43 (86%)	
Arterial hypertension	18 (37.5%)	32 (64%)	0.009
Osteoporosis/osteopenia	7 (14.6%)	21 (42%)	0.002
Visceral obesity	38 (79.1%)	44 (88%)	0.224
Metabolic syndrome	19 (39.6%)	29 (58%)	0.069
Hypercholesterolemia	14 (29.1%)	30 (60%)	0.002
Hypertriglyceridemia	11 (22.9%)	13 (26%)	0.486
Low HDL	14 (29.1%)	19 (38%)	0.361
Cardiovascular disease	0	5 (10%)	0.118
Peripheral vascular disease	0	1 (2%)	0.489
Diabetes mellitus	6 (12.5%)	24 (48%)	0.026
IFG	0	6 (12%)	0.622
IGT	6 (12.5%)	7 (14%)	0.678
IFG + IGT	1 (2%)	3 (6%)	0.457
Moon face	24 (50%)	33 (66%)	0.108
Myopathy	12 (25%)	36 (72%)	<0.001
Facial rubor	9 (18.7%)	23 (46%)	0.005
Buffalo hump	17 (35.4%)	33 (66%)	0.002
Purple striae	11 (22.9%)	15 (30%)	0.245
Hypovitaminosis D			
Deficiency	4 (8.4%)	26 (52%)	0.001
Insufficiency	10 (20.8%)	14 (28%)	0.545
Sufficiency	34 (70.8%)	10 (20%)	0.004

**Table 2 nutrients-14-00973-t002:** Anthropometric and biochemical parameters of patients with CD and controls at baseline.

	Controls Baseline (No. = 48)	Cushing’s Disease Baseline (No. = 50)	*p*
	Mean ± SD	Mean ± SD	
Age (yrs)	48.2 ± 13.4	50.9 ± 17.4	0.815
Anthropometric parameters			
BMI (kg/m^2^)	31.9 ± 5.01	33.1 ± 6.41	0.321
Waist circumference (cm)	105.4 ± 12.7	110.7 ± 8.97	0.031
Metabolic parameters			
Creatinine (mg/dL)	0.78 ± 0.25	0.81 ± 0.31	0.601
Calcium (mg/dL)	9.43 ± 0.46	9.46 ± 0.61	0.841
Phosphorus (mg/dL)	3.83 ± 0.67	3.46 ± 0.54	0.125
Parathyroid hormone (pg/mL)	33.8 ± 8.03	54.1 ± 22.7	0.003
25(OH)D (ng/mL)	28.7 ± 8.49	16.7 ± 8.18	<0.001
Glycaemia (mmol/L)	4.97 ± 2.77	6.66 ± 2.19	0.010
HbA1c (%)	5.79 ± 0.73	6.73 ± 1.09	0.004
Total cholesterol (mmol/L)	4.51 ± 0.82	5.34 ± 1.07	<0.001
HDL cholesterol (mmol/L)	1.15 ± 0.29	1.19 ± 0.45	0.184
Triglycerides (mmol/L)	1.66 ± 0.43	1.73 ± 0.67	0.585
LDL cholesterol (mmol/L)	2.62 ± 0.91	3.31 ± 0.99	0.002
HOMA-IR	3.07 ± 1.01	4.67 ± 2.83	0.051
ISI-Matsuda	4.14 ± 1.59	3.02 ± 2.18	0.007
Oral disposition index	3.75 ± 0.54	2.25 ± 2.04	0.003
Hormonal parameters			
ACTH (pmol/L)	7.72 ± 2.19	15.1 ± 6.56	<0.001
Mean urinary free cortisol (nmol/24 h)	310.2 ± 104.1	604.7 ± 65.6	0.001
Cortisol after low dose of dexamethasone suppression test (nmol/L)	44.4 ± 11.5	361.4 ± 98.4	0.001

**Table 3 nutrients-14-00973-t003:** Anthropometric and biochemical parameters at baseline and 6 weeks after cholecalciferol supplementation in patients with CD.

	Cushing’s Disease (No. = 50)		*p*
	Baseline	Six Weeks After Cholecalciferol	
	Mean ± SD	Mean ± SD	
Anthropometric parameters			
BMI (kg/m^2^)	33.1 ± 6.41	32.9 ± 7.43	0.880
Waist circumference (cm)	110.7 ± 8.97	109.8 ± 7.08	0.586
Metabolic parameters			
Creatinine (mg/dL)	0.81 ± 0.32	0.78 ± 0.26	0.615
Calcium (mg/dL)	9.46 ± 0.61	9.75 ± 0.56	0.017
Phosphorus (mg/dL)	3.46 ± 0.54	3.54 ± 0.43	0.424
Parathyroid hormone (pg/mL)	54.1 ± 22.7	40.5 ± 11.5	0.004
25(OH)D (ng/mL)	16.7 ± 8.18	30.7 ± 9.65	<0.001
Glycaemia (mmol/L)	6.66 ± 2.19	6.02 ± 1.65	0.109
Total cholesterol (mmol/L)	5.34 ± 1.07	4.87 ± 0.81	0.017
HDL cholesterol (mmol/L)	1.19 ± 0.45	1.21 ± 0.38	0.465
Triglycerides (mmol/L)	1.73 ± 0.67	1.68 ± 0.41	0.660
LDL cholesterol (mmol/L)	3.31 ± 0.99	2.98 ± 0.75	0.068
HOMA-IR	4.67 ± 2.83	3.97 ± 2.02	0.166
ISI-Matsuda	3.02 ± 2.18	3.76 ± 1.12	0.035
Oral disposition index	2.25 ± 2.04	2.97 ± 1.89	0.045
Hormonal parameters			
ACTH (pmol/L)	15.1 ± 6.56	14.3 ± 6.36	0.519
Mean urinary free cortisol (nmol/24 h)	604.7 ± 65.6	582.5 ± 54.9	0.075
Cortisol after low dose of dexamethasone suppression test (nmol/L)	361.4 ± 98.4	363.9 ± 89.6	0.895

**Table 4 nutrients-14-00973-t004:** Correlation of serum 25-hydroxyvitamin D [25(OH)D] levels at baseline in patients with Cushing’s disease and controls.

	25(OH)D			
	Cushing’s Disease		Controls	
	r	*p*	r	*p*
Glycaemia (mmol/L)	−0.385	0.019	−0.737	0.097
HbA1c (%)	−0.391	0.017	0.213	0.355
BMI (kg/m^2^)	−0.221	0.189	0.007	0.976
WC (cm)	−0.373	0.023	−0.130	0.042
ACTH (pmol/L)	−0.133	0.440	−0.198	0.567
Urinary free cortisol (nmol/24 h)	−0.466	0.033	0.040	0.862
Cortisol after low dose of dexamethasone suppression test (nmol/L)	−0.299	0.049	0.260	0.255

## Data Availability

Data are available on demand at corresponding author.

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
