# Peer review of "Vitamin D Deficiency in Cushing’s Disease: Before and After Its Supplementation"

_nutrients, 2022, doi:10.3390/nu14050973_

Round 1

Reviewer 1 Report

The use of simple methods to improve the condition of the body and reduce the effects of severe diseases is always an area of very valuable research. This is also the case here, except the key statistical analysis that was carried out. It should be emphasized that the authors collected considerable material, applied or explained the appropriate exclusion criteria, which at the design stage increases their value and reliability. They appropriately conducted the study that found native vitamin D deficiency in patients with CD. However, in the opinion of the reviewer, they did not properly perform the statistical analysis in the part concerning the response of the CD patients to supplementation with vitamin D (as below).

Introduction

Line 36:

Instead: 1,25-dihydroxyvitamin D3 or calcitriol; should: 1,25-dihydroxyvitamin D3 (calcitriol).

The record should preclude the interpretation that 1,25 (OH)2 D and calcitriol are two different compounds.

Line 49-50:

„In a clinical setting, controversial data have been reported with variable effects on plasma 1,25(OH)2D concentrations, including increases 50 [19], no change [20,21] and decreases [22]”.

What did the authors mean – variable effects of what?

Line 47-49: Short thought-you need to add a sentence about the function of the mentioned hydroxylases.

Generally, the introduction is too laconic and slogan - it should more precisely show the potential relationship of the functions of vitamin D with the biochemical and metabolic pathways in the CD pathomechanism. The authors should add a brief outline of the biochemical basis of the symptoms of CD, discussing the function of corticosteroids, effects on metabolism, mineral balance, cardiovascular system, etc..

Materials & Methods

Design of the study in the part concerning the response of the CD patients to supplementation with vitamin D is not appropriate for the intended purpose due to the use of the wrong baseline in statistical comparisons.

The authors used the comparison of CD group supplemented and control group (subjects not burdened with CD and with mostly sufficient in vitamin D level) supplemented, to demonstrate whether and how the supplementation of the vitamin D-deficient CD group affects the selected biochemical parameters characterizing CD.

Properly, the datasets analyzed should include: CD group after supplementation vs. CD group placebo with regard of CD placebo group vs. CD group at the starting point of the study. Supplementation of the control group is inappropriate because the reaction of the deficient organism (CD group) cannot be equated with the intake of high doses of vitamin D (150 000 I.U. for 6 weeks !) in the subjects with a normal concentration of 25 (OH) in the serum, in whom the body defends itself against overdosing by regulating mechanisms.

The only way to save the collected data and try to analyze it in terms of the overall effect of vitamin D supplementation on CD biochemical parameters is to analyze the data in the following system: CD group before supplementation vs. CD group after supplementation. This analysis will be more biased, but the view of vitamin D's effect will be more plausible. Only that when using such a system, the authors should very carefully formulate conclusions, rather showing possible trends in changes in parameters under the influence of supplementation.

On what basis was the dose of vitamin D - 150,000 IU selected?

Results

Table 1: The authors omitted osteoporosis/osteopenia as comorbidity characteristic for CD patients as compared with controls (p=0.002).

Discussion

Due to the fact that the statistical analysis of the results of the study of the effect of vitamin D supplementation on the biochemical parameters characterizing CD was questioned by the reviewer, the evaluation of the discussion in the current version is pointless.

General remarks

Not all abbreviations are explained with the first use in the text, such as: CS, WC, ADA criteria, NCEP ATP III criteria, ROC (explained only under Figure 2).

Author Response

The use of simple methods to improve the condition of the body and reduce the effects of severe diseases is always an area of very valuable research. This is also the case here, except the key statistical analysis that was carried out. It should be emphasized that the authors collected considerable material, applied or explained the appropriate exclusion criteria, which at the design stage increases their value and reliability. They appropriately conducted the study that found native vitamin D deficiency in patients with CD. However, in the opinion of the reviewer, they did not properly perform the statistical analysis in the part concerning the response of the CD patients to supplementation with vitamin D (as below).

Introduction

Line 36: Instead: 1,25-dihydroxyvitamin D3 or calcitriol; should: 1,25-dihydroxyvitamin D3 (calcitriol). The record should preclude the interpretation that 1,25 (OH)2 D and calcitriol are two different compounds.

Thanks for the comment. As you kindly suggested we changed the sentence.

Line 49-50: In a clinical setting, controversial data have been reported with variable effects on plasma 1,25(OH)2D concentrations, including increases 50 [19], no change [20,21] and decreases [22]”. What did the authors mean – variable effects of what?

Thanks for the comment. We clarified the sentence in the text.

Line 47-49: Short thought-you need to add a sentence about the function of the mentioned hydroxylases.

Thanks for the comment. As you suggested we added description of the hydroxylase function.

Generally, the introduction is too laconic and slogan - it should more precisely show the potential relationship of the functions of vitamin D with the biochemical and metabolic pathways in the CD pathomechanism. The authors should add a brief outline of the biochemical basis of the symptoms of CD, discussing the function of corticosteroids, effects on metabolism, mineral balance, cardiovascular system, etc..

Thanks for the comment. As you kindly suggested, we added a description of functions of vitamin D and of GCs effects.

Materials & Methods

Design of the study in the part concerning the response of the CD patients to supplementation with vitamin D is not appropriate for the intended purpose due to the use of the wrong baseline in statistical comparisons.

The authors used the comparison of CD group supplemented and control group (subjects not burdened with CD and with mostly sufficient in vitamin D level) supplemented, to demonstrate whether and how the supplementation of the vitamin D-deficient CD group affects the selected biochemical parameters characterizing CD.

Properly, the datasets analyzed should include: CD group after supplementation vs. CD group placebo with regard of CD placebo group vs. CD group at the starting point of the study. Supplementation of the control group is inappropriate because the reaction of the deficient organism (CD group) cannot be equated with the intake of high doses of vitamin D (150 000 I.U. for 6 weeks !) in the subjects with a normal concentration of 25 (OH) in the serum, in whom the body defends itself against overdosing by regulating mechanisms.

The only way to save the collected data and try to analyze it in terms of the overall effect of vitamin D supplementation on CD biochemical parameters is to analyze the data in the following system: CD group before supplementation vs. CD group after supplementation. This analysis will be more biased, but the view of vitamin D's effect will be more plausible. Only that when using such a system, the authors should very carefully formulate conclusions, rather showing possible trends in changes in parameters under the influence of supplementation.

Thanks for the proper comment. We deleted the data regarding the treatment of control group with cholecalciferol and we compared patients with CD before and after supplementation. We think that it may be interesting to compare at baseline controls and patients with CD before the supplementation of cholecalciferol.

On what basis was the dose of vitamin D - 150,000 IU selected?

Thanks for the question. Indeed, there are discordant recommendations on the oral load dose for cholecalciferol supplementation, We chose this dose according to the evidence that a dose over 100000 Units is able to determine a significant increase in serum/ plasma 25(OH)D concentration relative to baseline, which tended to peak between days 7 and 30 days. Notably, we based our search on some previous studies, such as “Effects of Three-Monthly Oral 150,000 IU Cholecalciferol Supplementation on Falls, Mobility, and Muscle Strengthin Older Postmenopausal Women: A RandomizedControlled Trial. J Bone Miner Res. 2012 Jan;27(1):170-6.doi: 10.1002/jbmr.524” “LARGE, SINGLE-DOSE, ORAL VITAMIN D SUPPLEMENTATION IN ADULT POPULATIONS: A SYSTEMATIC REVIEW. Endocr Pract. 2014 April ; 20(4): 341–351. doi:10.4158/EP13265.RA” who evaluated megadoses of cholcecalciferol supplementation in obese patients.

Results

Table 1: The authors omitted osteoporosis/osteopenia as comorbidity characteristic for CD patients as compared with controls (p=0.002).

Thanks for the comment. We added the lacking datum on osteoporosis/osteopenia in the text.

Discussion

Due to the fact that the statistical analysis of the results of the study of the effect of vitamin D supplementation on the biochemical parameters characterizing CD was questioned by the reviewer, the evaluation of the discussion in the current version is pointless.

General remarks

Not all abbreviations are explained with the first use in the text, such as: CS, WC, ADA criteria, NCEP ATP III criteria, ROC (explained only under Figure 2)

Thanks for the comment. We explained all the abbreviations, as you properly suggested.

Reviewer 2 Report

There are many published clinical studies that have assessed the prevalence of vitamin D deficiency in subjects with various diseases. Although good correlations are sometimes found between low vitamin D status and a disease, it has been difficult to determine whether the association is causative of the disease or is a consequence of the disease. In this study of patients with Cushing’s disease a good association was found between the severity of the disease, as assessed by mean urinary free cortisol output, and low concentrations of serum 25-hydroxyvitamin D, well illustrated in Figure 1. Two difficulties in this type of study, relate to two independent variables affecting vitamin D status. One of these is the seasonal variation in serum 25-hydroxyvitamin D concentrations, being higher in summer than in winter. The other, is that patients with a particular disease, may get less exposure to sunlight because of their ill-health, than healthy controls in the same environment.

In this study, the group of 50 subjects with Cushing’s disease was convened over a period of five years from 2016 to 2020. It is unclear whether the blood samples on which the 25-hydroxyvitamin D concentrations were determined were all collected at the same time of year during that 5-year period. If not, it is most important if the authors could divide the 50 subjects into two groups – those where the blood was collected over the months of summer and autumn, and those where blood was collected over the months of winter and spring. The question would then be was the vitamin D status still correlated with the disease severity and did the Cushing’s patients still have lower vitamin D status than the controls when blood samples from each group were assessed according to season?

Minor Points needing correction:

Line 20 and elsewhere: The terms 25(OH)D and other blood constituents are given values without indicating that these refer to serum concentrations. E.g. Line 19 should read “lower serum 25(OH)D concentrations” and Line 20 should read “had increased serum concentrations of both calcium (p=0.017) and 25(OH)D (p<0.001).” Elsewhere in the manuscript various assays are mentioned without indicating that the substances being assayed were from serum or some other fluid. Concentrations have no meaning unless the matrix from which values of the measured substance are given, are stated each time. So serum 25(OH)D concentration should be the form of words used, instead of simply “25(OH)D value”.

Line 30: It is incorrect to state that “vitamin D is a hormone”. The correct meaning here would be “Vitamin D is the precursor of a hormone with pleiotropic effects.”

Lines 49 and 55: The term “vitamin D levels” is used, referring to something measured in blood serum. This is incorrect. What was measured was the concentration of 25-hydroxyvitamin D, not vitamin D. Since both the parent molecule, vitamin D and 25-hydroxyvitamin D are present in serum samples, it is confusing to specify “vitamin D levels”, when in fact what was measured was 25-hydroxyvitamin D.

Author Response

There are many published clinical studies that have assessed the prevalence of vitamin D deficiency in subjects with various diseases. Although good correlations are sometimes found between low vitamin D status and a disease, it has been difficult to determine whether the association is causative of the disease or is a consequence of the disease. In this study of patients with Cushing’s disease a good association was found between the severity of the disease, as assessed by mean urinary free cortisol output, and low concentrations of serum 25-hydroxyvitamin D, well illustrated in Figure 1. Two difficulties in this type of study, relate to two independent variables affecting vitamin D status. One of these is the seasonal variation in serum 25-hydroxyvitamin D concentrations, being higher in summer than in winter. The other, is that patients with a particular disease, may get less exposure to sunlight because of their ill-health, than healthy controls in the same environment.

In this study, the group of 50 subjects with Cushing’s disease was convened over a period of five years from 2016 to 2020. It is unclear whether the blood samples on which the 25-hydroxyvitamin D concentrations were determined were all collected at the same time of year during that 5-year period. If not, it is most important if the authors could divide the 50 subjects into two groups – those where the blood was collected over the months of summer and autumn, and those where blood was collected over the months of winter and spring. The question would then be was the vitamin D status still correlated with the disease severity and did the Cushing’s patients still have lower vitamin D status than the controls when blood samples from each group were assessed according to season?

Thanks for the comment. To avoid seasonal influences, we only measured 25OHD levels in the months of winter and spring. We added it in the text.

Minor Points needing correction:

Line 20 and elsewhere: The terms 25(OH)D and other blood constituents are given values without indicating that these refer to serum concentrations. E.g. Line 19 should read “lower serum 25(OH)D concentrations” and Line 20 should read “had increased serum concentrations of both calcium (p=0.017) and 25(OH)D (p<0.001).” Elsewhere in the manuscript various assays are mentioned without indicating that the substances being assayed were from serum or some other fluid. Concentrations have no meaning unless the matrix from which values of the measured substance are given, are stated each time. So serum 25(OH)D concentration should be the form of words used, instead of simply “25(OH)D value”.

Thanks for the comment. As you properly suggested we added the source of assay for 25OHD.

Line 30: It is incorrect to state that “vitamin D is a hormone”. The correct meaning here would be “Vitamin D is the precursor of a hormone with pleiotropic effects.”

Thanks for the comment. We changed the sentence as you kindly suggested.

Lines 49 and 55: The term “vitamin D levels” is used, referring to something measured in blood serum. This is incorrect. What was measured was the concentration of 25-hydroxyvitamin D, not vitamin D. Since both the parent molecule, vitamin D and 25-hydroxyvitamin D are present in serum samples, it is confusing to specify “vitamin D levels”, when in fact what was measured was 25-hydroxyvitamin D.

Thanks for the comment. We changed the generic vitamin D with 25OHD in the text.

Round 2

Reviewer 1 Report

The authors took into account all comments of the reviewer. Statistical analysis is based on the suggested comparison of patients with CD before and after supplementation. The results interpreted correctly, though, as expected, are less spectacular compared to the first version of the analysis. Nevertheless, they conclusively indicate a positive effect of supplementation in patients with CD with vitamin D.

Attention is only drawn to the editing problem in the case of Table 3 - it requires corrective action.

Reviewer 2 Report

Thank you for the response to the comments of the reviewer.